# Diagnostic Potential for the Detection of Canine Visceral Leishmaniasis of an ELISA Assay Based on the Q5 Recombinant Protein: A Large-Scale and Comparative Evaluation Using Canine Sera with a Positive Diagnosis from the Dual-Path-Platform (DPP) Test

**DOI:** 10.3390/vetsci10100608

**Published:** 2023-10-07

**Authors:** Larissa Ferreira de Araújo Paz, Adalúcia da Silva, Hemilly Rayanne Ferreira da Silva, Milena Paiva Cavalcanti, Valeria Marçal Felix de Lima, Maria Rosário Oliveira da Cunha Beltrão, Maria Beatriz Araújo Silva, Osvaldo Pompílio de Melo Neto, Zulma Maria Medeiros, Wagner José Tenório dos Santos

**Affiliations:** 1Postgraduate Program in Health Sciences, University of Pernambuco, Recife 50100-010, Brazil; 2Aggeu Magalhães Institute—Fiocruz, Recife 50740-465, Brazil; 3Department of Clinic, Surgery and Animal Reproduction, College of Veterinary Medicine, São Paulo State University, Aracatuba 16050-680, Brazil

**Keywords:** *Leishmania infantum*, canine visceral leishmaniasis, DPP test

## Abstract

**Simple Summary:**

This study aimed to compare the true diagnostic potential of the recombinant chimeric protein Q5 in an ELISA assay using a large number of sera from CVL-suspected dogs. Sera from dogs with a CVL positive diagnosis based on the rapid DPP test (n = 406) and negative samples from healthy dogs (n = 46) were used for ELISA tests using the recombinant Q5. Overall, similar levels of lower sensitivity (67–68%) were seen for both the commercial EIE-LVC test and the Q5 ELISA when all assessed sera were considered, but a much greater sensitivity (92%) was seen for those samples from symptomatic dogs only. In contrast, many negative results were observed for the DPP-positive sera from asymptomatic dogs or from those with no clinical information available. The results reveal a higher-than-expected incidence of likely false-positive results for DPP, reinforcing the need for other recombinant proteins, such as the chimeric Q5, to be investigated as possible alternatives to the currently used CVL diagnostic methods.

**Abstract:**

Dogs are considered the major domestic reservoir for human visceral leishmaniasis, a serious disease caused by the *Leishmania infantum* parasite. Diagnosis of canine visceral leishmaniasis (CVL) is critical for disease control, with several methods currently available. Among the serological tests, the DPP rapid test and the EIE-LVC, more commonly used in Brazil, are associated with variable sensitivity and specificity. Research with novel recombinant proteins such as the ELISA with the recombinant chimeric protein Q5 may therefore improve the CVL diagnosis. This study aimed to evaluate the true diagnostic potential of Q5 in an ELISA assay using a large number of CVL-suspected sera (406) with a previous positive diagnosis based on the rapid DPP test. Sera from the DPP-positive dogs, also assessed with the EIE-LVC test, were compared with sera from healthy dogs (n = 46) and used for ELISA tests using the recombinant Q5. The resulting data as well as the correlation with the clinical signs and the environmental characteristics of the animals were analyzed using Medal and GraphPad Prism 8.0. Overall, similar levels of lower sensitivity (67–68%) were seen for both the commercial EIE-LVC test and the Q5 ELISA when all assessed sera were considered, but a much greater sensitivity (92%) was seen for those samples from symptomatic dogs only. In contrast, many negative results were observed for the DPP-positive sera from asymptomatic dogs or those with no clinical information available. A selection of those sera were tested yet again in new ELISA assays using a second batch of the recombinant Q5, purified under milder denaturing conditions, as well as using another recombinant protein (Lci13). The results reveal a higher-than-expected incidence of likely false-positive results for DPP, reinforcing the need for other recombinant proteins, such as the chimeric Q5, to be investigated as possible alternatives to the currently used CVL diagnostic methods.

## 1. Introduction

Domestic dogs are considered the main reservoir for visceral leishmaniasis (VL), having a fundamental role in urban areas for the human transmission of *Leishmania infantum*, the protozoan parasite responsible for most cases of the disease in Latin America and southern Europe. Indeed, positive prevalence in the canine population can reach up to 80% in highly endemic areas, with various studies suggesting that there is an overlap between locations with an incidence of human cases and high canine seroprevalence [1,2,3]. The number of infected dogs with *L. infantum* in South America is estimated to be in the millions, with greater evidence of disease growth in Brazil, Argentina and Paraguay [4]. In endemic areas of VL caused by *L. infantum*, such as Brazil, infected dogs are characterized by a large accumulation of parasites. The rapid and efficient detection of the infection in animals with and without clinical signs is therefore essential to control the spread of the disease to other dogs and humans, with the high rate of asymptomatic animals being a significant complication [5,6].

Several methods are considered for the VL diagnosis, with both parasitological and molecular tests having operational limitations for canine visceral leishmaniasis (CVL). Many serological techniques are commercially available and in use, but many of these are based on complex preparations of native proteins derived from the parasite, limiting the specificity and sensitivity of the tests and facilitating false-positive results due to cross-reactions with antigens from other pathogens, such as *Babesia* and *Ehrlichia.* At the same time, a greater incidence of false-negative cases may favor the spread of the disease through misdiagnoses, limiting the identification of asymptomatic animals [5,6,7,8]. The official serological diagnosis of CVL in Brazil uses the rapid DPP (Dual Path Platform) test, consisting of a device impregnated with the recombinant rK28 antigen, a chimera combining three *L. infantum* antigens (K9, K26 and K39) [9,10]. Another recommended test is EIE-LVC, an ELISA assay based on soluble antigens from *Leishmania major* promastigote forms [11]. A study using these two serological tests (DPP and EIE-LVC) revealed that only 67.2% of positive samples based on real-time PCR (polymerase chain reaction) were positive in both DPP and EIE [12]. Yet another study assessing the DPP test, but with sera from CVL-positive animals confirmed through culture or PCR, also found low sensitivity (74%) and specificity (94%) values, although these were still higher than those seen for the confirmatory test (EIE-LVC), with 67% sensitivity and 87% specificity [7]. Co-infection with pathogens such as *Babesia* sp. and *Ehrlichia* sp. in dogs from urban areas endemic or not endemic for CVL was shown to interfere with the serological diagnosis using ELISA, IFAT (immunofluorescence antibody test) and DPP tests [13].

Diagnostic limitations in asymptomatic animals and cross-reactions with other infections from trypanosomatides have spurred the quest for novel recombinant antigens. Different studies have been carried out aiming to improve the quality of individual recombinant antigens or combinations of antigens applied to the diagnosis of CVL [14]. Our group has previously identified the recombinant Lci13, a fragment of the mitochondrial 70 kDa heat-shock protein from *L. infantum* [15], as potentially effective for the CVL diagnosis (97% sensitivity), but not for the diagnosis of the human disease [16]. A second recombinant protein evaluated by our group, the chimeric protein Q5, was designed to join fragments derived from *L. infantum* recombinant antigens previously found to be efficient for either the human or canine forms of the disease (Lci2, Lci3 and Lci12). Preliminary results with a limited set of 39 sera from infected dogs with a positive parasitological and/or molecular diagnosis showed a 99% sensitivity for Q5, with 100% specificity defined after testing 15 sera from healthy animals [17]. Given the promising results with Q5, the present study was carried out in order to better evaluate the Q5 performance in an ELISA assay designed for the CVL serodiagnosis in dogs, using a much larger set of canine sera (from symptomatic and asymptomatic dogs) previously assessed with the two tests currently prescribed for CVL diagnosis in Brazil, DPP and EIE-LVC. DPP-positive sera having negative results for the Q5 ELISA and/or EIE-LVC were further tested with a second Q5-based ELISA assay as well as yet another ELISA based on a different recombinant protein, Lci13. The comparative analyses confirm equivalent performances for both Q5 ELISA and EIE-LVC for symptomatic animals only while revealing a large number of DPP-positive sera with consistent negative results for all ELISA assays tested, mainly from asymptomatic dogs or from animals with no defined clinical symptoms.

## 2. Materials and Methods

### 2.1. Canine Sera and Ethical Considerations

The canine sera assayed here were provided by CVL epidemiological surveillance teams (responsible for carrying out confirmatory VL tests) from the state of Pernambuco, Northeastern Brazil. Information on reported epidemiological and clinical conditions was collected at the time of registration of the VL-positive dogs. All sera considered suspicious for CVL, as based on a positive rapid DPP test result (n = 406), were from Pernambuco. Sera from healthy dogs (n = 48) were used as negative controls, with 30 of those from the State of São Paulo (non-endemic area) and 18 from the State of Pernambuco (endemic area), all with negative results for CVL seen using the DPP test, as well as a previously described PCR assay [18]. The research started after evaluation and approval by the Ethics Committee on the Use of Animals CEUA/UPE under registration No. 03/2020.

### 2.2. Protein Expression and Purification

Expression and purification of the recombinant Q5 used for the first set of ELISA assays (8 M urea batch) was carried out as previously described [17], using the Q5 gene cloned into the pRSET expression (Invitrogen, Waltham, MA, USA) transformed into *Escherichia coli* Rosetta^TM^ 2 DE3 (from Novagen, Darmstadt, Germany). The recombinant Q5 construct and the one encoding the Lci13 antigen, which was also cloned into the pRESTa plasmid [16], were then used in a second purification procedure to produce the 2 M urea batches. These used bacterial cells from one-liter cultures which were harvested and resuspended in 20 mL of buffer A (50 mM Tris-HCl pH 8.0, 300 mM NaCl, 20 mM imidazole, 2 M urea, 10 mM 2-Mercaptoethanol) and lysed with five pulses of ultrasonication at 4 °C. Soluble supernatants, after centrifugation at 20,000× *g* for 30 min at 4 °C, were then loaded onto a 5 mL His-Trap HP column in an AKTA Pure system (Cytivia, Marlborough, MA, USA) equilibrated with buffer A. Proteins were eluted with a two-step gradient in a 10 column volume (CV) linear gradient from 0% to 10% buffer B (buffer A + 500 mM imidazole), followed by a 20 CV linear gradient from 10% to 100% buffer B. Elutions were loaded on 15% SDS-PAGE followed by Coomassie staining to confirm the protein load and quality [1].

### 2.3. Diagnostic Tests and ELISA

The tests DPP and the EIE-LVC, both produced at Bio-Manguinhos (Fiocruz, Rio de Janeiro, Brazil) for CVL diagnosis, were used as recommended by the manufacturer. The ELISA assays with the recombinant Q5 and Lci13 were performed essentially as described previously [17], using 600 ηg per well of the recombinant proteins. For the assays, the wells were incubated with the canine sera at dilutions of 1:900 (for Q5) or 1:200 (for Lci13), using as the second antibody the peroxidase-conjugated goat anti-dog IgG (diluted 1:10,000, from Sigma Aldrich (St. Louis, MO, USA), catalog no. A6792 (Q5 2M and Lci13) and diluted 1:1200, from Jackson Immuno Research 304-005-003 for Q5 8M). For the ELISA assays with recombinant proteins, sera were used in triplicates. Sera dilutions were defined using a standardization curve ranging from 1:100 to 1:900. To assess the clinical performance of the assay, the Q5 was evaluated in retrospective clinical specimens to compare its diagnostic capability with reference methods. A total of 454 sera were tested with DPP, with all DPP-positive sera also tested subsequently with the EIE-LVC assay. Figure 1 details the number of samples tested with each of the ELISA assays evaluated here.

### 2.4. Data Collection

After the interview with the pet owner, a physical examination of the dog was conducted and peripheral venous blood was collected from the dog. The samples were stored, processed, and analyzed at the municipality (physical examination and a rapid DPP test), and other diagnostics were performed in Fiocruz Pernambuco.

### 2.5. Statistical Analyses

The data were tabulated in Microsoft Excel 2013 and the sensitivity, specificity, positive predictive value, negative predictive value, accuracy and confidence interval parameters were estimated using the MedCalc website. Scatterplots, ROC curve, Kappa Index and AUC were generated using GraphPad Prism 8.0.2. Concordance assessment between tests was conducted using a kappa statistic, with the results interpreted as follows: values ≤ 0 indicated no agreement; 0.01–0.20 meant none to slight agreement; 0.21–0.40, fair; 0.41–0.60, moderate; 0.61–0.80, substantial; and 0.81–1.00 indicated an almost-perfect agreement. Cut-off for the ELISA assays was determined by calculating the average of the values derived from the negative sera plus twice the standard deviation.

## 3. Results

### 3.1. Environmental and Clinical Characteristics of the Study Population

To properly evaluate the Q5 recombinant protein in a large-scale setting, we assembled a substantial set of canine sera using dogs from an area endemic for CVL with a suspected diagnosis for the disease confirmed using the recommended DPP assay (n = 406). The animals whose sera were evaluated here consisted predominantly of domestic dogs (60.1%), mostly males (52.7%), and the dogs were generally one year old or older (67.5%) and known to have had some type of contact with humans (69%). Regarding their sheltering conditions, 24% were classified as intra-residence (found within the households), 70% as peri-residence (sheltered within an area equal to or less than 100 m from a residence), and 51% as extra-residence (sheltered within a radius greater than 100 m from a residence). Sera from asymptomatic animals represented 28.8% of those tested, with 11.8% of the sera derived from symptomatic animals; there was no clinical information available for the remaining dogs (59.4%). Considering only the symptomatic dogs, the main symptoms were onychogryphosis (47.9%), generalized dermatitis (72.9%) and weight loss (45.8%). Table 1 summarizes all the clinical and environmental data available for the DPP-positive dogs included in our analysis.

### 3.2. Large-Scale Evaluation of the Recombinant Q5 for the CVL Diagnosis

To assess the efficiency of Q5 for the CVL diagnosis with the larger assemblage of DPP-positive sera, ELISA assays were set up as previously described [17], using the recombinant protein purified in the presence of 8 M urea (Q5-8M). When the whole set of tested sera was considered, the performance of the Q5 ELISA was lower than expected, with an overall sensitivity of 68% and a specificity of 94% (Figure 2A). However, when the Q5-8M performance was assessed with only the sera from dogs with defined clinical features, either symptomatic or asymptomatic, the sera from the symptomatic dogs were identified with a substantially greater sensitivity (92%) than were the asymptomatic sera (62%). ROC curves were then generated for these various groups of sera (Figure 2B), and all were seen to have AUC values close to 1.0 (0.852, 0.951 and 0.835, respectively, for the total, symptomatic and asymptomatic groups, with *p* < 0.0001). Sensitivity, specificity and accuracy values for the sera groups assessed are also shown in Table 2.

### 3.3. Comparison between EIE-LVC and Q5

According to the standard protocol for current CVL diagnosis in Brazil, all canine sera found to be positive using the DPP assay were also tested with the EIE-LVC assay prior to their use in the experiments described here. To better assess the Q5-8M performance, we therefore opted to directly compare the results from the ELISA assay with those derived from the EIE-LVC testing. The results, also summarized in Table 2, show an overall sensitivity of 67% when all DPP-positive sera are considered, but with 92% and 58% sensitivities, respectively, for the sera from symptomatic and asymptomatic dogs. The specificity was not determined for EIE-LVC, since this assay was only applied to sera previously tested with DPP, and none of these included samples from healthy-control animals. We then calculated the agreement between the two tests and that between both assays and the results determined with DPP. Identical values were observed for both Q5 and the EIE-LVC (0.48 for both), indicating a moderate agreement through the kappa index. In contrast, a greater agreement was seen between the Q5-8M ELISA and EIE-LVC (0.65), indicating a substantial agreement between these two ELISA assays. The limited agreement seen between both ELISA assays and DPP led us to investigate in more detail the positive and negative results seen for each test with each serum. Overall, and considering the whole set of DPP-positive sera (n = 406), 272 sera were identified as CVL-positive and 134 as negative for the EIE-LVC test, while 275 were positive and there were 131 negatives for the Q5-8M ELISA. Remarkably, 101 sera were negative for both assays despite having a previous DPP positive result. Only three of those were from dogs with clinical symptoms associated with CVL, contrasting with 38 sera from asymptomatic dogs and 60 from animals with no clinical information available.

### 3.4. Comparative Evaluation of Different Q5-Based ELISA Assays

The large number of sera which were found to be positive for the DPP assay and negative for the Q5 ELISA might reflect a lower efficiency of the Q5 recombinant protein in identifying the asymptomatic animals or those with lower antibody levels. We reasoned that one contributing factor might be the purification conditions used for the Q5 production, highly denaturing in the presence of 8 M urea, a condition originally chosen to improve the solubilization of the recombinant protein and maximize yield during the purification steps. To consider this, we opted to evaluate a new batch of recombinant Q5, which was prepared using a new purification protocol under milder denaturing conditions (2 M urea—Q5-2M), and to assess a selection of the DPP-positive sera which were mostly negative for both EIE-LVC and the previous Q5-8M ELISA. Figure 3A compares the purification yield of both protocols used for the Q5 purification using SDS-PAGE, with both protocols leading to efficient protein purifications, but with a higher yield for the 2 M urea purification. We then set up a new ELISA assay with the Q5-2M protein, opting to also change the secondary antibody used in the assay, again aiming to see if further chances could improve sensitivity. Due to limitations in serum availability, the new Q5-2M ELISA was then evaluated with a selection of the DPP-positive sera which mostly included those which were negative for both EIE-LVC and the previous Q5-8M ELISA. A total of 169 DPP-positive sera, mostly from asymptomatic animals or with no information available regarding clinical symptoms, were selected for the new evaluation, with 126 of those having a negative result with the first Q5-8M ELISA experiment (the results are summarized in Table 3). In all, 43 sera were found to be positive with the new assay, with only 23 coinciding with a previous positive result for the Q5-8M ELISA. A total of 109 sera remained negative in both assays; however, only 88 were also found to be negative with the EIE-LVC test, including 85 sera from asymptomatic animals or those with no symptomatology available.

### 3.5. Evaluation of the Lci13 Recombinant Protein

Since the number of negative results for the DPP-positive sera remained substantial even after the second Q5 ELISA, we also considered evaluating a selection of the negative sera with yet another ELISA assay. For this second assay, we chose the Lci13 recombinant protein, previously shown to have an excellent performance for CVL diagnosis [16]. Lci13 is based on a fragment of the *L. infantum* mitochondrial HSP70 heat-shock protein [15] and shares no elements in common with the recombinant antigens which are the basis for the DPP and the Q5 ELISA assays. We then assessed a total of 119 DPP-positive sera using the Lci13 ELISA, with 85 of those having previous negative results for both Q5 ELISA assays as well as the EIE-LVC (results are summarized in Figure 4 and Appendix A). Negative results were seen for 101 of the sera tested. From the remaining 18 positive sera, nine were also found to be positive with the new Q5 ELISA and six with both sets of Q5 based tests. In all, 76 DPP-positive sera were found to produce negative results for all tests evaluated here. Most of these (73) were from asymptomatic dogs or from animals with no information regarding their clinical symptoms, but they constituted ~18% of all DPP-positive sera used in this investigation. A single serum from a healthy control animal from the endemic area was also found to be positive with all three ELISA assays tested here.

## 4. Discussion

The dual-path rapid chromatographic immunoassay (DPP) is based on the rK28 antigen, which is a recombinant chimeric antigen protein derived from the fusion of fragments from both the *L. donovani* K39 and haspb1 antigens with the entire haspb2 sequence [10]. DPP was described as effective for the CVL diagnosis and is currently recommended by the Brazilian Ministry of Health to be used as a first assay for CVL identification, with the EIE-LVC ELISA used for confirmatory purposes. Here, we opted to follow the current Brazilian guidelines and thus used the criteria of DPP positivity for the selection of the sera to be used in this study. DPP, however, has been recently used to investigate CVL incidence through serological surveys, in which was found a lower-than-ideal efficiency with asymptomatic animals [20,21,22]. Further limitations of DPP regarding the detection of CVL in asymptomatic dogs were found to be associated with a higher-than-expected incidence of false-negative [7] or both false-negative and false-positive results [12]. Our results further revealed a possibly relevant incidence of false-positive results associated with this test, especially considering the large-scale setting, a finding which must be considered in future assessments of the performance of different diagnostic tests.

Both rK28 and Q5 are chimeric proteins derived from the fusion of fragments from three *Leishmania* antigens, with the rK39 and Lci2 antigens, found, respectively, within rK28 and Q5, being both derived from the same native *Leishmania* antigen, a N-type kinesin [9,23]. Similar performances for rK28 and Q5 might have been expected considering their common features, but this was not seen to be the case here, with a large number of DPP-positive sera from asymptomatic animals and those with no known symptomatology having negative results with the Q5 ELISA. An association between clinical signs and positivity might be due to seroconversion, mainly because symptomatic dogs have higher levels of anti-*Leishmania* spp. antibodies [8]. Serological tests may thus be less efficient in detecting VL infections with asymptomatic dogs [24,25]. Indeed, it has been shown that some recombinant antigens may have a better efficacy than others in confirming the CVL diagnosis in sera from asymptomatic dogs [22], and this possibility was also raised by us to explain these differences in cases where Q5 was inefficient for the diagnosis of the asymptomatic animals. However, the results from the EIE-LVC test were mostly in agreement with those seen for the Q5 ELISA. The EIE-LVC test is based on a complex mixture of multiple antigens [11], which would be more consistent with a higher degree of positivity than would any test made with individual recombinant proteins. These results are supported by a previous report, with a reduced number of sera evaluated, where the number of DPP-positive sera was also substantially greater than those seen with other assays using complex antigenic mixtures, EIE-LVC and IFAT [13].

A third ELISA assay tested here, using the recombinant Lci13, produced results in agreement with both the Q5 ELISA and the EIE-LVC. Lci13 shares no elements in common with either DPP or Q5 and, on a previous evaluation, it was seen to be an excellent antigen for CVL diagnosis using the ELISA assay [16]. At this stage, it is not possible to rule out the possibility that the differences in performance observed might also be impacted by the different methods used, namely, rapid test or ELISA. Nevertheless, the significant number of DPP-positive sera which were found here to have negative results with the EIE-LVC test as well as both the Q5 and Lci13 ELISA assays raises concerns regarding the use of DPP for the diagnosis of asymptomatic animals, suggesting a substantial number of false-positive results. These results are consistent with a previous comparative analysis in which several DPP-positive results were seen from samples where a real-time PCR assay did not find evidence of *Leishmania* DNA [12].

In this study, we used stored serum samples from routine CVL epidemiological surveillance, which could be seen as a potential limitation, as we cannot infer anything about the clinical outcomes of the dogs. Moreover, since we did not explore the presence of other opportunistic infections in the dogs evaluated, this could act as a confounder factor in serum-diagnosed CVL. Further cohort studies could clarify the potential influence of other opportunistic and latent coinfections.

The pilot study in which the recombinant Q5 chimera was first described evaluated a total of 39 CVL positive sera, with the diagnosis confirmed either by PCR and/or culture, using as control sera from healthy young animals from a non-endemic area. The ELISA assays from these sera showed values of 100% sensitivity and specificity with Q5, indicating a very good performance for the CVL diagnosis [17]. These results are markedly different from the ones seen in the current study when the whole set of DPP-positive sera is considered.

## 5. Conclusions

The discrepancies seen here with the ELISA results for the sera from asymptomatic animals and those with no known clinical symptoms do not allow the inclusion of these groups for a proper evaluation of the Q5 performance, for the reasons discussed above. When only the sera from symptomatic animals are considered, the results from the previous pilot study are confirmed by the data from the current study, reinforcing the excellent potential for the use of the Q5 protein, or improved versions, for the CVL diagnosis, either in ELISA assays, or as part of rapid tests. Such tests can be included as part of proper control strategies aiming to increase the early CVL diagnosis in dogs and reduce the spread of the disease, including appropriate treatment for infected animals, use of repellent collars and vector-control measures to reduce transmission.

## Figures and Tables

**Figure 1 vetsci-10-00608-f001:**
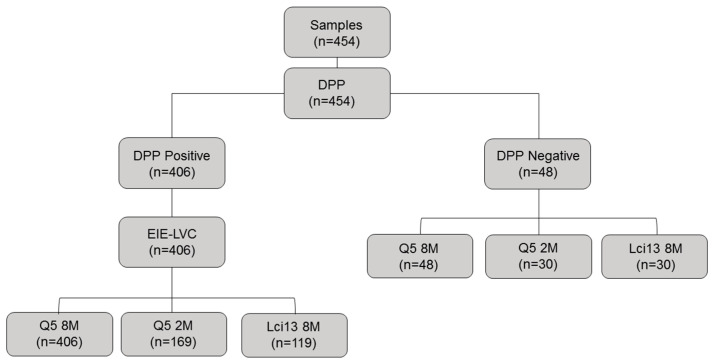
Flowchart detailing the number of samples tested in each test. DPP-screened positive samples were tested with EIE-LVC to confirm the diagnosis of CVL. Samples from healthy and DPP-negative animals were used as a negative control and later tested with a lab-on-chip assay to evaluate the diagnostic methodology.

**Figure 2 vetsci-10-00608-f002:**
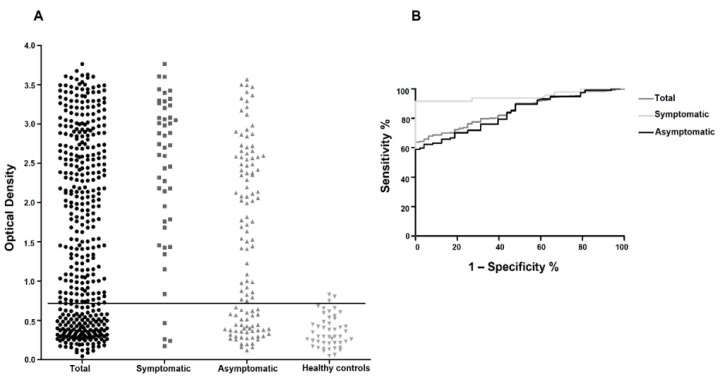
Evaluation of the recombinant Q5-8M with a large set of sera from VL-positive dogs. (**A**) ELISA results with the recombinant Q5 from the testing of the 406 DPP-positive sera, as well as the symptomatic (48 sera), asymptomatic (117 sera) and healthy-control groups (48 sera). (**B**) ROC curves for the three DPP-positive groups generated with the results shown in (**A**).

**Figure 3 vetsci-10-00608-f003:**
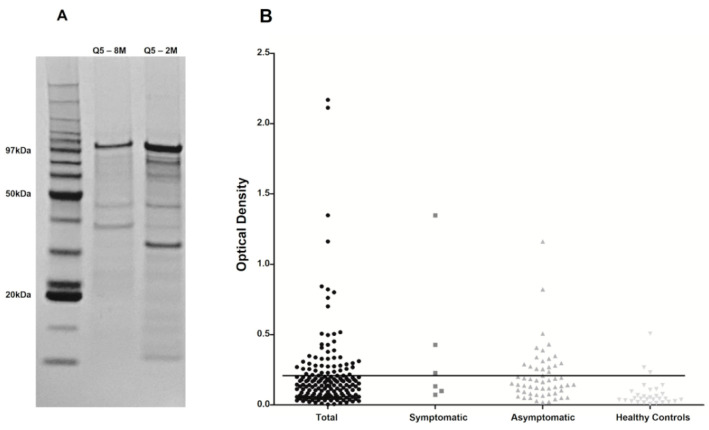
Comparative evaluation of a second batch of the Q5 chimeric protein. (**A**) SDS-PAGE gel showing the different protein purifications of Q5 using 8 M and 2 M urea. (**B**) ELISA results with the recombinant Q5 purified with 2 M urea and tested with 169 DPP-positive (Total), Symptomatic and Asymptomatic sera as well 30 control sera (Healthy Controls).

**Figure 4 vetsci-10-00608-f004:**
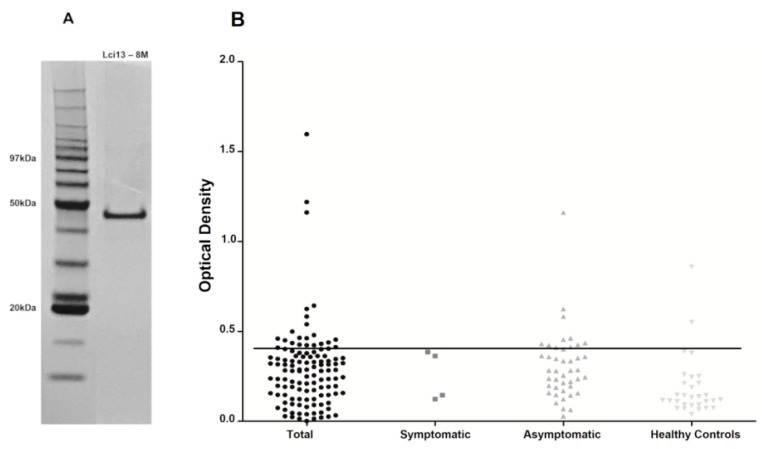
Evaluation of the Lci13 recombinant protein for the CVL diagnosis. (**A**) SDS-PAGE gel showing the purified recombinant Lci13 used for the ELISA assays. (**B**) ELISA results with the selected 119 DPP positive (Total), Symptomatic, and Asymptomatic sera as well as the 30 control sera (Healthy Controls).

**Table 1 vetsci-10-00608-t001:** Distribution of clinical and environmental characteristics of the animals whose sera were evaluated here.

N = 406	Characteristics	N (%)
Origin of the animal	Domiciled	244 (60.1)
	Semi-domiciled	4 (1.0)
	Communitarian	63 (15.5)
	Errant	6 (1.5)
	NI	89 (21.9)
Sex	Female	176 (43.3)
	Male	214 (52.7)
	NI	16 (3.9)
Age	Up to 12 months	21 (5.2)
	One year old or older	274 (67.5)
	NI	111 (27.3)
Human contact	Yes	280 (69.0)
	No	2 (0.5)
	NI	124 (30.5)
Type of shelter *	Intra-residence	100 (24.6)
	Extra-residence	51 (12.6)
	Peri-residence	70 (17.2)
	NI	185 (45.6)
Symptoms	Asymptomatic	117 (28.8)
	Symptomatic	48 (11.8)
	NI	241 (59.4)
Clinical characteristics	Generalized dermatitis	35 (72.91)
	Onychogryphosis	23 (47.91)
	Weight loss	22 (45.83)
	Hair loss	13 (27.08)
	Keratoconjunctivitis	7 (14.58)

* [19]. NI: No information available.

**Table 2 vetsci-10-00608-t002:** Evaluation of the sensitivity, specificity, accuracy and kappa index of the CVL Q5 test with the Q5 chimeric protein, in comparison with the EIE-LVC, and assessing the whole set of the DPP positive sera evaluated as well as only those from the symptomatic and asymptomatic groups.

ELISA	Sensitivity % (n = 406)	Symptomatic (n = 48)	Asymptomatic (n = 117)	Specificity % (n = 48)	Accuracy %	Kappa
Q5-8M	68 (63–72)	92 (80–98)	62 (53–71)	94 (83–99)	71 (66–75)	0.48 (0.38–0.60)
EIE-LVC	67 (61–72)	92 (80–98)	58 (50–68)	Not assayed	Not assayed	0.48 (0.40–0.60)

All results were calculated with a 95% confidence interval (CI).

**Table 3 vetsci-10-00608-t003:** Comparison of ELISAs Q5 (8M), Q5 (2M) and EIE-LVC, showing the number of positive and negative samples derived from each test.

	Q5 (8 M) Positive	Q5 (8 M) Negative	Q5 (8 M) Positive	Q5 (8 M) Negative
	Q5 (2 M) Positive	Q5 (2 M) Positive	Q5 (2 M) Negative	Q5 (2 M) Negative
EIE-LVC positive	4	7	4	21
EIE-LVC negative	19	13	13	88
Total	23	20	17	109

Q5 (8M) and Q5 (2M)—ELISA tests were carried out with the Q5 recombinant protein purified using 8-molar and 2-molar urea, respectively.

## Data Availability

Raw data are available from the corresponding author, and are also provided in Appendix A.

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
