# Peer review of "Diagnostic Potential for the Detection of Canine Visceral Leishmaniasis of an ELISA Assay Based on the Q5 Recombinant Protein: A Large-Scale and Comparative Evaluation Using Canine Sera with a Positive Diagnosis from the Dual-Path-Platform (DPP) Test"

_vetsci, 2023, doi:10.3390/vetsci10100608_

Round 1
Reviewer 1 Report
Line 69: “cross-reactions with other diseases” try to be more specific, the cross-reactions is not with other diseases but with pathogen's antigen
Line 69 “euthanasia” It is not necessary could be enough a treatment and a protection with insecticide that don’t allow sand flies biting activity
Line 70-72: “the occurrence of false-negative cases may also favor the spread of the disease through misdiagnosis can also have limitations for the diagnosis of asymptomatic animals” protection of all negative dogs with insecticide and repellent treatments (collar, spot-on) is strongly recommended, as a control measure, should be mentioned
Line 78: Space missing after reference [12]
Line 86: “have stimulated the search for new recombinant antigens” please modify the reported sentence, perhaps it could be better to specify a more appropriate verb.
Line 91: “by us” please reconsider the sentence trying not to use those words
Line 108: “was” bold formatting error
Line 109: “All sera considered positive, based on the rapid DPP test (n=406), were from Pernambuco” positive control should not be based only on a serological technique (DPP rapid test) in the absence of the demonstration of the parasite presence (microscopy, PCR, culture) the control group could not be considered as positive. The same standard methods should be applied as for negative sera.
Line 17-173: “with a confirmed positive diagnosis using the recommended DDP assay (n=406)” as mentioned above this sentence should be changed positive sera are not confirmed by a non-immunoenzimatic test
Table 2: Please consider to divide the first row to asses only numbers reported in the first row and Q5 values for the second row as for EIE-LVC
Results presented are adequates.
English language could be improved, but only minor editing are required
Author Response
Thank you very much for your comments and please see the attachment.

Reviewer 2 Report
The manuscript talks about an important topic and the results are good. However, the manuscript needs to go through a through English revision. There are many grammatical mistakes and unclear sentences. I have attached the pdf version with comments and highlighted some of the errors in yellow. The title need to be improved. I have put a suggestion in the corrected pdf. Section 2.2 (Methods) SDS -PAGE needs to be included. Conclusion should also be improved.

The manuscript need to be revised for gramatical errors and clarity
Round 2
Reviewer 1 Report
The modified manuscript is appropriate and suitable for publication
Author Response
Thank you very much once again for the considerations.